# The Triglyceride/High-Density Lipoprotein Cholesterol (TG/HDL-C) Ratio as a Risk Marker for Metabolic Syndrome and Cardiovascular Disease

**DOI:** 10.3390/diagnostics13050929

**Published:** 2023-03-01

**Authors:** Constantine E. Kosmas, Shanna Rodriguez Polanco, Maria D. Bousvarou, Evangelia J. Papakonstantinou, Edilberto Peña Genao, Eliscer Guzman, Christina E. Kostara

**Affiliations:** 1Division of Cardiology, Department of Medicine, Montefiore Medical Center, Bronx, NY 10467, USA; 2Cardiology Clinic, Cardiology Unlimited, PC, New York, NY 10033, USA; 3School of Medicine, University of Crete, 710 03 Heraklion, Greece; 4General Directorate of Public Health and Social Welfare, Attica Region, 115 21 Athens, Greece; 5Laboratory of Clinical Chemistry, School of Health Sciences, Faculty of Medicine, University of Ioannina, 451 10 Ioannina, Greece

**Keywords:** triglycerides, HDL-C, TG/HDL-C ratio, risk marker, metabolic syndrome, cardiovascular disease, coronary artery disease, peripheral artery disease, cerebrovascular disease

## Abstract

Atherosclerosis is an immunoinflammatory pathological procedure in which lipid plaques are formed in the vessel walls, partially or completely occluding the lumen, and is accountable for atherosclerotic cardiovascular disease (ASCVD). ACSVD consists of three components: coronary artery disease (CAD), peripheral vascular disease (PAD) and cerebrovascular disease (CCVD). A disturbed lipid metabolism and the subsequent dyslipidemia significantly contribute to the formation of plaques, with low-density lipoprotein cholesterol (LDL-C) being the main responsible factor. Nonetheless, even when LDL-C is well regulated, mainly with statin therapy, a residual risk for CVD still occurs, and it is attributable to the disturbances of other lipid components, namely triglycerides (TG) and high-density lipoprotein cholesterol (HDL-C). Increased plasma TG and decreased HDL-C levels have been associated with metabolic syndrome (MetS) and CVD, and their ratio, TG/HDL-C, has been proposed as a novel biomarker for predicting the risk of both clinical entities. Under these terms, this review will present and discuss the current scientific and clinical data linking the TG/HDL-C ratio with the presence of MetS and CVD, including CAD, PAD and CCVD, in an effort to prove the value of the TG/HDL-C ratio as a valuable predictor for each aspect of CVD.

## 1. Introduction

The pathophysiology of atherosclerosis depends vastly on lipid transportation along with inflammation.

Inflammation plays a crucial role in the pathogenesis of cardiovascular disease (CVD), as it has been linked with both the initiation and progression of atherosclerosis [1,2]. Several pro-inflammatory cytokines, such as the C-reactive protein (CRP), tumor necrosis factor-α (TNF-α) and interleukin-6 (IL-6), have been unequivocally shown to promote both insulin resistance (IR) and atherogenesis [1,2,3]. Furthermore, inflammation plays a major role in the pathogenesis of chronic lung disease [4], as well as in the pathogenesis of chronic kidney disease (CKD), also including the development of malnutrition–inflammation–atherosclerosis syndrome (MIA), which is one of the causes of increased mortality in CKD [3,5]

The formation and subsequent potential rupture of unstable atherosclerotic plaques is mediated by disruptions in the physiology of lipid metabolism. Macrophages take up and accumulate oxidized triglycerides (oxTGs) and cholesterol particles in their cytoplasm, a procedure that effectuates their transformation into lipid-enriched cells named foam cells. The latter are the cornerstone in atherogenesis, which precedes atherosclerotic cardiovascular disease (ASCVD). Coronary artery disease (CAD), peripheral vascular disease (PAD) and cerebrovascular disease (CCVD) constitute the three aspects of ASCVD. Hence, defects in lipid metabolism induce atherogenesis and an increase in the risk of CVD [6]. According to WHO, 32% of global deaths in 2019 were attributed to CVD; CAD and CVA were held responsible for 85% of them [7].

Ample clinical evidence suggests a strong association between elevated low-density lipoprotein cholesterol (LDL-C) and poor cardiovascular outcomes [8]. Statins are the standard-of-care for managing LDL-C levels. Aggressive lowering of LDL-C, even beyond the previously established cut-off target of 70 mg/dl, has been demonstrated to ameliorate cardiovascular outcomes [9,10,11,12,13]. Nevertheless, a noteworthy subset of patients remains in peril for CVD events, in spite of optimal LDL-C control [14]. This residual metabolic risk could be attributed to various atherogenic processes that continue to exist even after aggressive LDL-C lowering and is plausibly correlated with certain novel biomarkers, addressing new data concerning the optimal holistic approach of this specific subset of patients [15]. An association between increased plasma triglyceride (TG) and decreased high-density lipoprotein cholesterol (HDL-C) levels with metabolic syndrome (MetS) and CVD (including CAD, PAD and CCVD) has been proposed. More specifically, a descriptive parameter known as triglyceride/high-density lipoprotein cholesterol (TG/HDL-C) ratio has been strongly correlated with insulin resistance (IR) and central obesity, both of them being aspects of the MetS, which can enhance the risk of CVD [16]. HDL-C has been linked to cardioprotective effects via its antioxidant and anti-inflammatory properties [6]. The current literature supports that decreased HDL-C levels predispose to CVD, although interventions targeting an increase in HDL-C levels per se have not been proven successful in decreasing the risk of CVD [17]. It is well known that atherosclerosis is a significant CVD-related mortality factor in patients with end-stage renal disease, as it appears to be about 10 to 30 times more prevalent compared to the general population. A retrospective study including 973 patients in peritoneal dialysis (PD) concluded that a higher serum TG/HDL-C ratio was an independent variable in terms of predicting all-cause and CVD mortality in young and older PD patients [18].

Bearing in mind the existing evidence, the TG/HDL-C ratio could arise as a promising marker for the assessment of CVD risk, morbidity and mortality and may become a valuable tool in terms of addressing strategies for primary and secondary prevention [19].

## 2. Triglyceride/HDL-C Ratio in Metabolic Syndrome

The National Cholesterol Education Program Adult Treatment Panel III (ATP III) defines metabolic syndrome as central obesity (waist circumference: >102 cm in men and >88 cm in women), abnormal lipid panel (HDL-C <40 mg/dl in men and <50 mg/dl in women and TG ≥150 mg/dl), elevated blood pressure (systolic blood pressure [SBP] ≥130 mmHg or diastolic blood pressure [DBP] ≥85 mmHg) and insulin resistance/glucose intolerance (fasting glucose ≥100 mg/dl). The diagnosis is clear when at least three of these criteria are met [20,21]. The worldwide MetS prevalence is estimated to be from 20 to 25% in the adult population [21]. MetS augments the likelihood of CVD, diabetes mellitus (DM) and chronic kidney disease (CKD) [22]. Whilst clinicians have established a connection between elevated serum levels of LDL-C and CVD, the identification of biomarkers applicable for MetS and its active part in atherogenesis may be challenging [23]. For the past decade, extensive research has been conducted regarding the interlink between TGs and other lipoproteins, yet has not been entirely fruitful. Recent studies have demonstrated that the TG/HDL-C ratio is a convenient tool for detecting IR, one of the main components of MetS. This tool is practical and a potential alternative to the conventional insulin assay, allowing an early identification of the cardiometabolic hazards before the advent of clinical manifestations and complications [24,25,26].

Scientific research data support the positive correlation between MetS, CVD and the value of the TG/HDL-C ratio. An early analysis by Mc Laughin et al. in 2003 focused on the relation between TG/HDL-C ratio and cardiometabolic risk at a cut-off point of 3.0 units, for both men and women. A total of 258 overweight volunteers (body mass index, [BMI] ≥25 kg/m^2^) with no previous diagnosis of hypertension or diabetes were included. Plasma TG levels, TG/HDL-C ratio and insulin concentration were measured. Data concluded that the use of TG/HDL-C ratio in overweight patients could be a valuable tool for identifying insulin resistance in those patients, which poses them at an increased risk for CVD [25].

In another recent cross-sectional study involving over 5000 Iranian participants, anthropometric measures and blood pressure were taken and the patients were categorized according to their lipid ratios (total cholesterol/HDL-C ratio, LDL-C/HDL-C ratio and TG/HDL-C ratio). After adjusting for various variables (age, gender, body mass index and past medical history), the researchers concluded that the TG/HDL-C was the best indicator for identifying metabolic syndrome compared to the other ratios [22]. Another study in Iran demonstrated that the high TG/HDL-C ratio was associated with a 2.12 times increased risk of developing metabolic syndrome, using a cut-off point of 4.03 for males and 2.86 for females [27].

In 2021, a cross-sectional study was conducted in the elderly Chinese population, which included a total of 1267 participants ≥65 years of age. The purpose for the researchers was to investigate a correlation between TG/HDL-C ratio and MetS. They determined that TG/HDL-C ratio values exceeding the cut-off values of 1.437 for men and 1.196 for women predicted a higher risk of developing MetS [28]. Additionally, the Korean National Health and Nutrition Examination Survey conducted a large-scale study concerning the TG/HDL-C ratio and its relationship with metabolic syndrome. The mean TG/HDL-C ratio increased along with the number of MetS components. The cut-off point of the TG/HDL-C ratio for the fourth quartile was 3.52 and, after adjustment, the odds ratio (OR) for MetS in the fourth quartile compared with that of the first quartile was 29.65 in men and 20.60 in women (*P* < 0.001) [29].

Last but not least, a multicentered study in Brazil enrolled 2472 multiethnic participants free of major cardiovascular risk factors and defined the TG/HDL-C ratio cut-off value of 2.6 for men and 1.7 for women. The results of this study demonstrated that these cut-off values were reliable and showed good clinical applicability to detect cardiometabolic disorders. Moreover, these cut-off values demonstrated great sensitivity and specificity regardless of the ethnicity or age of the participants, although the black race showed lower values of the TG/HDL-C ratio, compared with other ethnic groups [26].

Undoubtedly, the TG/HDL-C ratio is a very satisfactory predictor for MetS. Nonetheless, taking into consideration the different cut-off values of multiple trials, based on ethnicity, genetics and lifestyle, the aforementioned ratio cannot be considered an absolute parameter without calibration. The cumulative risk factors are well established through the different studies; therefore, the TG/HDL-C ratio could function as an atherogenic index for MetS [30,31,32].

A summary of the results of the clinical studies relating to the TG/HDL-C ratio and Metabolic Syndrome is shown in Table 1.

## 3. Triglyceride/HDL-C Ratio in Coronary Artery Disease

Coronary artery disease (CAD) is defined as lipid accumulation and the formation of atherosclerotic plaques below the tunica intima of the coronary arteries causing a narrowing and, consequently, a partial or total vessel obstruction. According to the Center for Disease Control and Prevention (CDC), it is the most common type of heart disease [33]. Multiple methods, such as nuclear magnetic resonance spectroscopy, non-denaturing gradient gel electrophoresis and density gradient ultracentrifugation, are available as accurate markers of lipid panels that can be associated with the extent of coronary disease visualized in coronary angiography [34,35,36], although their high cost and necessity for technical expertise render them impractical. Instead, researchers have focused on discovering more accessible atherogenic indices that would predict the risk of developing CAD using laboratory tests for assessment of lipid profiles [35]. Several studies have stated that high serum TG levels and low HDL-C are associated with an increased risk of both acute coronary syndrome (ACS) and stable CAD even when desirable levels of LDL-C have been achieved. This statement reveals that, despite optimization of total cholesterol (TC) and LDL-C levels, a residual risk for cardiovascular events persists [37].

In a study conducted in a geriatric Chinese population, it was demonstrated that the TG/HDL-C ratio was an excellent predictor of major adverse cardiovascular events (MACE) in patients with ACS who underwent percutaneous coronary intervention (PCI). A total of 1694 patients treated with primary or elective PCI were enrolled and were followed for a median follow-up interval of 31 months. Findings revealed that the TG/HDL-C ratio, as well as other TG-derived metabolic indices, were all strongly associated with and were excellent predictors of MACE among ACS patients undergoing PCI [37].

In another recent study in a Chinese population, it was shown that the TG/HDL-C ratio was also a good predictor of the presence of coronary artery calcification [38].

The TG/HDL-C ratio also appears to have a role in the setting of non-obstructing coronary artery disease (NOCAD). A study published by Prasad et al. enrolled 465 patients with reported signs and symptoms indicative of ischemic cardiovascular disease. Subsequent imaging via coronary angiography did not reveal any stenosis >20%. The patients were followed for 7.8 ± 4.3 years for the development of MACE. The results of this study showed that a high TG/HDL-C ratio was associated with an increased risk of MACE in postmenopausal women, whereas no such association was observed in men with NOCAD [39].

Similar to coronary angiography, coronary CT angiography (cCTA) is a useful, non-invasive tool for assessing the characteristics of coronary plaques. A study in Okayama University Hospital in Japan in 2020 included 944 patients with suspected stable CAD who underwent cCTA. The population was classified into two groups using values of ≥2.0 and <2.0 for the TG/HDL-C ratio. A comparison between these two groups and high-risk plaque characteristics, such as low attenuation (<50 HU), positive remodeling (remodeling index >1.1) and spotty calcification, was taken into consideration. The results demonstrated that the prevalence of plaques, and especially the high-risk, vulnerable plaques, was significantly higher in the group with the higher TG/HDL-C ratio, thus supporting a prognostic value of TG/HDL-C ratio for potential future cardiovascular events [40].

Su YM et al. conducted a study which concluded that a high TG/HDL-C ratio is a predicting factor for repeated revascularization. They enrolled patients who had an ACS and underwent a successful PCI and showed that an increased TG/HDL-C ratio was an independent risk factor for recurrent PCI. This was more evident in male patients and especially those with a history of arterial hypertension and hyperglycemia [41].

A study conducted in 482 Australian patients with a follow-up period of 5.1 years showed that a TG/HDL-C ratio >2.5 is an independent predictor of long-term all-cause mortality and is strongly associated with an increased risk of MACE [42].

Finally, Kundi et al. studied 1342 patients who underwent PCI. They had either typical angina symptoms and/or a positive treadmill or myocardial perfusion scintigraphy test. The study used a cut-off point of 3.8 for the pre-procedural TG/HDL-C ratio. Its sensitivity and specificity were 71% and 68%, respectively, with regard to the prediction of in-stent restenosis (ISR). Furthermore, patients in the highest quartile of TG/HDL-C ratio had the highest rate of ISR [43]. The authors suggested that a high TG/HDL-C ratio might increase the risk of ISR via increased insulin resistance, endothelial dysfunction and atherosclerosis, oxidative stress, pro-inflammatory status, and proliferation of vascular smooth muscle cells. Notwithstanding, further studies are needed to definitely assess the role of TG/HDL-C ratio as a risk marker for ISR.

It has also been reported that a higher TG/HDL-C ratio is linked to the presence of small-dense LDL particles, which are highly atherogenic. As it has been noted above, the TG/HDL-C ratio has been recognized as a reliable risk marker of ASCVD. Thus, the above data support the application of the TG/HDL-C ratio as a management target for minimizing residual risk and eventually preventing future atherogenic events [44].

A summary of the results of the clinical studies relating to TG/HDL-C ratio and coronary artery disease is shown in Table 2.

## 4. Triglyceride/HDL-C Ratio in Peripheral Artery Disease

Peripheral artery disease (PAD) constitutes blood flow restriction in the arteries of the extremities, most commonly in the lower than in the upper ones, attributed to atherosclerosis [45]. PAD is strongly associated with other conditions such as CAD and CVD [46]. In vivo invasive catheter-based imaging techniques, such as intravascular ultrasound (IVUS), can distinguish the different composition of plaques in PAD and CAD, so as to address the underlying risk of both clinical conditions. In accord with these studies, one-half to two-thirds of plaques in CAD are composed of lipid cores, while plaques in PAD have been described as fibrocalcific, with lipid cores being present in only 25% of them [47].

Aday et al. followed 27,888 women from the Women’s Health Study (WHS) for a median of 15.1 years. Measurements of standard lipid concentrations, as well as nuclear magnetic resonance (NMR) lipoprotein profiling were performed. NMR-derived measures of LDL particle concentration (but not LDL-C levels) were associated with incident PAD. Moreover, other features of atherogenic dyslipidemia, including elevations in TC/HDL-C ratio, elevations in triglyceride-rich lipoproteins and low standard and NMR-derived measures of HDL were also significant risk predictors. This study raised the belief that identification of specific dyslipidemia-related risk markers that would reliably predict the risk for PAD may have clinical benefits in patients at risk for PAD for whom limited medical therapy options exist to date [48]. Thus, further studies are needed in order to establish clinically reliable risk markers that would predict the risk for PAD.

Ding et al. performed a study in a Chinese population sample of 10,900 hypertensive patients, who were given a diagnosis of PAD based on an ankle-brachial index (ABI) < 0.9. The relationship between nonconventional lipid markers and the presence of PAD was studied in order to improve the risk identification and management. The TG/HDL-C ratio was strongly associated with the risk of PAD with a 14% increase in the risk of PAD per one standard deviation (SD) increment of the TG/HDL-C ratio. Moreover, the adjusted OR of the TG/HDL-C ratio for PAD was 1.71 when comparing the highest tertile (TG/HDL-C ratio of ≥ 1.3) to the lowest tertile (TG/HDL-C ratio of < 0.7) [49]. In addition, the Atherosclerosis Risk in Communities or ARIC study followed 8330 black and white individuals, free of PAD at baseline. Out of these, 246 developed PAD over a median follow-up interval of 17 years. The results revealed an independent and robust association between higher baseline levels of TG-related blood lipids and lower levels of HDL-related lipids with incident PAD. On the contrary, there was lack of statistically significant associations of TC and LDL-C with PAD [50].

In a cross-sectional study, which included 314 diabetic patients with symptoms suggestive of PAD, it was shown that the mean TG/HDL-C ratio was significantly higher in patients with PAD, as compared to normal controls (7.88 vs. 4.98, respectively, in men and 8.06 vs. 5.98, respectively, in women; *p* = 0.03) [51].

Furthermore, the TG/HDL-C ratio appears to also be predictive of the complexity of PAD. This was demonstrated in a study, which included 412 patients with lower limb claudication who underwent peripheral angiographic procedures and they were assigned to two groups based on angiographic lesion complexity, according to the TransAtlantic Inter-Society Consensus-II (TASC II) classification. Group 1 comprised those with TASC A-B lesions (314 patients) and group 2 comprised those with TASC C-D lesions (98 patients). Both groups had similar baseline characteristics in terms of age, BMI and chronic diseases. The results demonstrated that the TG/HDL-C ratio was significantly higher in group 2 (patients with lesions of higher angiographic complexity). A TG/HDL-C ratio cut-off value of 2.9 predicted angiographic complexity with a sensitivity and specificity of 75.5% and 56.7%, respectively. Thus, this study proved that the TG/HDL-C ratio could reliably predict the complexity of PAD [52].

A universal atherogenic index capable of predicting the risk for PAD would facilitate prevention and intervention strategies for this clinical entity. Evidence towards predictability of PAD via biomarkers is encouraging but is still far than sufficient, although recent studies have set the foundation for further research by luminously describing the complexity of this relationship.

A summary of the results of the clinical studies relating to the TG/HDL-C ratio and peripheral artery disease is shown in Table 3.

## 5. Triglyceride/HDL-C Ratio in Cerebrovascular Disease

The American Heart Association (AHA)/American Stroke Association (ASA) defines stroke as brain injury that can be attributable to ischemic (atherothrombotic plaque) or embolic or hemorrhagic causes (ruptured blood vessels) [53,54]. Evidence suggests a stronger association of lipoproteins with ischemic stroke compared with hemorrhagic stroke and the ratio of apoB/A1 has been reported to be the best lipid predictor of ischemic stroke risk [55]. Regardless of the type of stroke, lipid assessment serves as an important tool to determine metabolic disorders that may represent a potential risk for recurrent stroke, disability and/or death [56].

There is ample evidence supporting the association of TG and HDL-C with stroke. A total of 8072 out of 267,500 participants in six Chinese cohort studies were diagnosed with a stroke. Elevated serum TG levels correlated positively with ischemic stroke but not with hemorrhagic stroke. However, the risk of both types of stroke was enhanced with low HDL-C levels [57]. On the other hand, another Chinese trial enrolling 4995 individuals with hypertension found no relation between TG and HDL-C variability and ischemic stroke [58].

The predictive value of major lipids for stroke severity and outcome was investigated in a trial with 790 participants with stroke. Stroke severity was assessed according to the National Institutes of Health Stroke Scale (NIHSS), with a score ≥ 5 indicating moderate-to-severe stroke. Data indicated that lower TG and HDL-C levels were associated with more severe stroke, whereas lower TG levels also appeared to predict in-hospital mortality [59].

A study, which followed 3216 American Indian participants for a median of 17.7 years, showed that non-diabetic patients with high TG (≥ 150 mg/dl) and low HDL-C levels (< 40 mg/dL for men and < 50 mg/dL for women) had a 2.13-fold greater hazard ratio (HR) for stroke [60].

In a Chinese study, 3172 healthy participants were enrolled. The researchers evaluated patients who showed up for medical check-ups. Silent brain infarcts (SBI) were diagnosed in participants who were asymptomatic with >3 mm diameter lesions on T1- or T2- weighted images. A total of 263 patients (8.3%) had SBI. The TG/HDL-C ratio was independently associated with SBI (adjusted odds ratio [aOR] = 1.16; *p* = 0.047). This association was robust in males (aOR = 1.23; *p* = 0.021), but not in females. Moreover, the TG/HDL-C ratio was positively correlated with an SBI lesion burden in a dose-response manner (*P* for trend = 0.015) [61].

In another study, which included 851 neurologically healthy participants, it was shown that there was a positive association between high TG/HDL-C ratio and the development of intracranial atherosclerosis (ICAS). Subjects with ICAS were significantly more likely to have a high tertile TG/HDL-C ratio (>2.06) than a low tertile TG/HDL-C ratio (< 1.37) after adjusting for cardiovascular risk factors (OR = 1.83; *p* = 0.03) [62].

Furthermore, in another study, which included 112 young patients with ischemic stroke (mean age of 38.46 years), 113 healthy adults and 110 patients with ischemic stroke aged >45 years (mean age of 69.53 years), it was shown that the TG/HDL-C ratio was significantly higher in young patients with stroke than in healthy adults. Moreover, the TG/HDL-C ratio was also significantly higher in young patients with stroke than in older cases [63].

On the other hand, there is evidence that a higher TG/HDL-C ratio may predict a better prognosis in patients with acute ischemic stroke (AIS). In a study which included 1006 Chinese patients with AIS, the effect of the TG/HDL-C ratio on 3-month mortality after AIS was assessed. The results revealed that the prognosis of the group of patients with a TG/HDL-C ratio > 0.9 was markedly superior to that of the group with a TG/HDL-C ratio ≤ 0.9 (*P* < 0.001). In addition, a multivariate Cox regression analysis showed that the TG/HDL-C ratio was independently correlated with a reduced risk of mortality (HR = 0.39; *P* < 0.001) [64]. In another study by the same group, which included a total of 1981 patients with AIS, it was determined that a lower TG/HDL-C ratio was significantly associated with a higher risk of hemorrhagic transformation (HT) in patients with AIS attributable to large artery atherosclerosis (LAA) [OR = 0.53; *p* = 0.032], but not in patients with cardioembolism or small-vessel occlusion [65].

Nonetheless, further large studies are needed to definitely assess the role of the TG/HDL-C ratio as a predicting marker of the risk of stroke and its prognosis.

Ultimately, stroke is one of the leading causes of disability and death worldwide. Therefore, the ultimate goal in the prevention and management of this disease should be directed towards identification of biochemical disturbances that play an important role in its pathophysiology and pathogenesis. Thus, the search for novel biomarkers, such as the TG/HDL-C ratio, intends to identify a risk indicator of cerebrovascular disease with the currently available data [66].

A summary of the results of the clinical studies relating to the TG/HDL-C ratio and cerebrovascular disease is shown in Table 4.

## 6. Conclusions

Taking everything into account, disturbances in lipid metabolism are responsible for atherogenesis and a subsequent increased prevalence of ASCVD, including coronary artery disease, peripheral artery disease and cerebrovascular disease. The development of CVD may still occur in spite of the strict regulation of LDL-C with statins, indicating the presence of residual risk, which may be attributable to other lipid elements. Metabolic syndrome is another risk factor for cardiovascular disease and is proven to be correlated with increased plasma triglyceride and decreased HDL-C levels. The TG/HDL-C ratio is closely related to insulin resistance and central obesity, and thus to metabolic syndrome. The current literature contains a plethora of trials and studies establishing the TG/HDL-C ratio as an excellent novel risk marker, suitable for predicting the risk for MetS, CAD, PAD, and CCVD, although the absence of a universal cut-off value renders its use as a predicting biomarker for CVD somewhat challenging.

In a study published in the e-journal of the European Society of Cardiology (ESC) Council for Cardiology Practice, TG/HDL-C ratio values >2.75 in men and >1.65 in women were found to be highly predictive of MetS, as well as of a first coronary event regardless of BMI [67]. In another elegant study, which was designed to compare discrimination for cardiovascular risk by different cut-off values of the TG/HDL-C ratio, receiver operating characteristic (ROC) analysis was performed for the relationship between TG/HDL-C ratio and accumulation of cardiometabolic risk factors, including visceral obesity, hypertension and DM. The optimal cut-off values for TG/HDL-C ratio in ROC analysis were 2.967 in men and 2.237 in women, providing odds ratios for multiple cardiometabolic risk factors of subjects with vs. subjects without a high TG/HDL-C ratio of 5.75 in men and 18.76 in women [31].

In another important study, which used log-binomial regression analyses and ROC curve analysis to analyze associations of serum concentrations for fasting insulin, TG and HDL-C among 2652 nondiabetic adults in the United States, it was again clearly demonstrated that the TG/HDL-C ratio may be a clinically simple and useful indicator for IR among nondiabetic adults regardless of race/ethnicity. Importantly though, it also became evident that a single optimal cutoff point may not be applicable across diverse racial/ethnic subpopulations. More specifically, the results of this study showed that the optimal cutoff point of the TG/HDL-C ratio for the prediction of hyperinsulinemia was 3.0 for non-Hispanic whites and Mexican Americans and 2.0 for non-Hispanic blacks [68].

The above data ought to set the foundation for the future exploring of the best possible cut-off values, sensitivity and specificity of the TG/HDL-C ratio across diverse racial/ethnic subgroups, so as to more effectively add this ratio into the diagnostic armamentarium of CVD.

In addition, further studies are needed to establish other new lipid-related biomarkers and to compare the reliability of these biomarkers, or even different combinations of them, in terms of their clinical efficacy in predicting the risk for MetS and CVD. Moreover, relevant comparison of the new lipid-related biomarkers with the TG/HDL-C ratio would be important. This may also be useful for establishing guidelines for the prevention of CVD in clinical practice.

Notwithstanding, the above should not take away the fact that the TG/HLC-C ratio has been proven to be an excellent novel risk marker for effectively predicting the risk for MetS and CVD.

## Figures and Tables

**Table 1 diagnostics-13-00929-t001:** Summary of the results of the clinical studies relating to TG/HDL/C ratio and metabolic syndrome.

Study	Design	Method	Results
The Atherogenic Index Log (Triglyceride/HDL-Cholesterol) as a Biomarker to Identify Type 2 Diabetes Patients with Poor Glycemic Control [16]	Prospective cohort	TG/HDL-C ratio measurement	The log (TG/HDL-C) can be considered as a biomarker to predict T2D patients with poor glycemic control. The best cut-off point of log (TG/HDL-C) for the discrimination between patients with HbA1c ≥8% versus patients with HbA1c <8% determined to be 0.44.
Comparison of Lipid Ratios to Identify Metabolic Syndrome [22]	Cross-sectional	TC/HDL-C, TG/HDL-C and LDL/HDL-C ratio	The results suggest that TG/HDL-C ratio is a better marker for identifying MetS in the Iranian population.
Use of metabolic markers to identify insulin resistant overweight individuals [25]	Cross-sectional	TG, TG/HDL-C and insulin concentration	Cut-point of TG/HDL-C ratio was 3.0 for both male and female participants. The sensitivity and specificity were 64% and 68%, respectively, regarding identification of insulin resistance and diagnosis of MetS.
Reference values for the triglyceride to high-density lipoprotein ratio and its association with cardiometabolic diseases in a mixed adult population: The ELSA-Brazil [26]	Prospective cohort	Anthropometric measurement and TG/HDL-C ratio	Cut-off values of TG/HDL-C ratio were 2.6 for males and 1.7 for females, displaying great sensitivity and specificity, regardless of the ethnicity or age.
Lipid ratio as a suitable tool to identify individuals with MetS risk: A case- control study [27]	Case–control	Serum lipids and MetS criteria	High TG/HDL-C ratio increases by 2.12 times the possibility of having MetS. Its cut-off points were 4.03 for men and 2.86 for women.
High TG/HDL ratio suggests a higher risk of metabolic syndrome among an elderly Chinese population: a cross-sectional study [28]	Cross-sectional	Anthropometric parameters and blood drawn for lipid panel	TG/HDL-C ratio values exceeding the cut-off values of 1.437 for men and 1.196 for women predicted a higher risk of developing MetS.
The Relationship between the Triglyceride to High-Density Lipoprotein Cholesterol Ratio and Metabolic Syndrome [29]	Cross-sectional	Anthropometric measurements and TG/HDL-C ratio	The cut-off point of the TG/HDL-C ratio for the fourth quartile was 3.52 and, after adjustment, the OR for MetS in the fourth quartile compared with that of the first quartile was 29.65 in men and 20.60 in women (*P* < 0.001).

**Table 2 diagnostics-13-00929-t002:** Summary of the results of the clinical studies relating to TG/HDL-C ratio and coronary artery disease.

Study	Design	Method	Results
Impact of non-fasting triglycerides/high-density lipoprotein cholesterol ratio on secondary prevention in patients treated with statins [19]	Prospective	Routine blood tests before and after PCI; follow-up of 5 years	Patients in the 2nd and 3rd quantile (TG/HDL-C ratio of 2.6 and 5.6, respectively) had higher incidence of major adverse cardiovascular events, despite having LDL-C levels < 100 mg/dl.
Prognostic significance of multiple triglycerides-derived metabolic indices in patients with acute coronary syndrome [37]	Single-center, prospective cohort	Venous blood sample and percutaneous coronary intervention; median follow-up of 927 days (927–1109 days)	The TG/HDL-C ratio has a strong relationship with the risk of MACE in patients with ACS who underwent PCI.
Triglyceride to High-Density Lipoprotein Ratio can predict coronary artery calcification [38]	Retrospective case–control	Coronary angiography to determine the presence of coronary artery calcifications (CACs) and laboratory testing	The TG/HDL-C ratio is a good predictor for the presence of CACs. The diagnostic threshold was 1.037, and the corresponding sensitivity and specificity were 89.3% and 60.5%, respectively.
Triglyceride and Triglyceride/HDL (High Density Lipoprotein) Ratio Predict Major Adverse Cardiovascular Outcomes in Women With Non-Obstructive Coronary Artery Disease [39]	Prospective cohort	Coronary angiography and laboratory testing	Postmenopausal women with NOCAD and high TG/HDL-C ratio have increased risk of MACE.
Triglyceride to HDL-Cholesterol ratio is a predictor of future coronary events: a possible role of high-risk coronary plaques detected by coronary CT angiography [40]	Prospective cohort	Coronary CT angiography and bloodwork	Patients with TG/HDL-C ratio ≥2.0 had a higher prevalence of high-risk plaques in cCTA.
Triglyceride to high-density lipoprotein cholesterol ratio as a risk factor of repeat revascularization among patients with acute coronary syndrome after first-time percutaneous coronary intervention [41]	Prospective cohort	Percutaneous coronary intervention and blood work	TG/HDL-C ratio was an independent predictor of repeat PCI.
Elevated Triglycerides to High-Density Lipoprotein Cholesterol (TG/HDL-C) Ratio Predicts Long-Term Mortality in High-Risk Patients [42]	Single-center, prospective cohort. Follow-up period of 5.1 years.	Blood sample for lipid panel collected before undergoing coronary angiography	A TG/HDL-C ratio >2.5 is an independent predictor of long-term all-cause mortality and is strongly associated with an increased risk of MACE.
Is In-Stent Restenosis After a Successful Coronary Stent Implantation Due to Stable Angina Associated With TG/HDL-C Ratio? [43]	Retrospective	Treadmill test or myocardial scintigraphy test; percutaneous coronary angiography and blood sample	A TG/HDL-C ratio cut-off value of 3.8 predicted in-stent restenosis with a sensitivity and specificity of 71% and 68%, respectively.

**Table 3 diagnostics-13-00929-t003:** Summary of the results of the clinical studies relating to TG/HDL-C ratio and peripheral artery disease.

Study	Design	Method	Results
Arterial function parameters in patients with metabolic syndrome and severe hypertriglyceridemia [21]	Prospective cohort	Medical questionnaire, physical examination (height, weight, waist circumference, blood pressure), pulse wave velocity (PWV), electrocardiogram, endothelium-dependent flow-mediated dilation test, carotid artery ultrasound	Patients with MetS had increased carotid arterial stiffness.
Lipoprotein particle profiles, standard lipids, and peripheral artery disease incidence: prospective data from the Women’s Health Study [48]	Prospective cohort	Plasma cholesterol and triglyceride concentration and measured lipoprotein particles using a proton nuclear magnetic resonance (NMR) spectroscopy	NMR-derived measures of LDL particle concentration (but not LDL-C levels) were associated with incident PAD. Elevations in TC/HDL-C ratio, elevations in triglyceride-rich lipoproteins, and low standard and NMR-derived measures of HDL were also significant risk predictors.
Association between nontraditional lipid profiles and peripheral arterial disease in Chinese adults with hypertension [49]	Cross-sectional	ABI and blood lipid panel	The TG/HDL-C ratio is strongly associated with the risk of PAD with a 14% increase in the risk of PAD per SD increment of the TG/HDL-C ratio.
Conventional and Novel Lipid Measures and Risk of Peripheral Artery Disease [50]	Prospective cohort	Fasting lipids, symptoms and medical history	There is an independent and robust association between higher baseline levels of TG-related blood lipids and lower levels of HDL-related lipids with incident PAD.
The role of triglycerides and triglyceride/high-density lipoprotein ratio as a positive predictive factor in peripheral vascular disease [51]	Cross-sectional	Demographic data and routine blood tests were obtained	In patients with diabetes, the mean TG/HDL-C ratio was significantly higher in those with PAD, as compared to normal controls (7.88 vs. 4.98, respectively, in men and 8.06 vs. 5.98, respectively, in women; *p* = 0.03).
Is it possible to predict the complexity of peripheral artery disease with atherogenic index? [52]	Retrospective cohort	Lipid panel and angiographic procedures	A TG/HDL-C ratio cut-off value of 2.9 predicted angiographic complexity with a sensitivity and specificity of 75.5% and 56.7%, respectively.

**Table 4 diagnostics-13-00929-t004:** Summary of the results of the clinical studies relating to TG/HDL-C ratio and cerebrovascular disease.

Study	Design	Method	Results
Association of Lipids, Lipoproteins, and Apolipoproteins with Stroke Subtypes in an International Case Control Study (INTERSTROKE) [55]	Case–control	Lipid levels measurement	A stronger association of lipoproteins with ischemic stroke compared with hemorrhagic stroke was noted. The ratio of apoB/A1 was the best lipid predictor of ischemic stroke risk.
Association of Lipids With Ischemic and Hemorrhagic Stroke: A Prospective Cohort Study Among 267,500 Chinese [57]	Prospective cohort	Lipid measurement	High TG values are related to ischemic but not hemorrhagic stroke. Low HDL-C values are related to increased risk of both ischemic and hemorrhagic stroke
The association of blood lipid parameters variability with ischemic stroke in hypertensive patients [58]	Retrospective cohort	Lipid measurement	No relation was found between TG and HDL-C variability with ischemic stroke.
Prognostic significance of major lipids in patients with ischemic stroke [59]	Prospective cohort	Brain computed tomography and lipid assessment	Lower TG and HDL-C levels were associated with more severe stroke, whereas lower TG levels also appeared to predict in-hospital mortality.
Triglyceride and HDL-C Dyslipidemia and Risks of Coronary Heart Disease and Ischemic Stroke by Glycemic Dysregulation Status: The Strong Heart Study [60]	Prospective cohort	Metabolic panel	Non-diabetic patients with high TG (≥ 150 mg/dl) and low HDL-C levels (< 40 mg/dL for men and < 50 mg/dL for women) had a 2.13-fold greater HR for stroke.
High triglyceride/HDL cholesterol ratio is associated with silent brain infarcts (SBI) in a healthy population [61]	Retrospective	T1- or T2- weighted images and lipid panel	The TG/HDL-C ratio was independently associated with SBI (adjusted odds ratio [aOR] = 1.16; *p* = 0.047). This association was robust in males (aOR = 1.23; *p* = 0.021), but not in females. Moreover, the TG/HDL-C ratio was positively correlated with SBI lesion burden in a dose-response manner (*P* for trend = 0.015).
Triglyceride/HDL-Cholesterol Ratio as an Index of Intracranial Atherosclerosis (ICAS) in Nonstroke Individuals [62]	Retrospective	Measurements of serum lipids, brain magnetic resonance imaging (MRI) and magnetic resonance angiography (MRA)	Subjects with ICAS were significantly more likely to have a high tertile TG/HDL-C ratio (> 2.06) than a low tertile TG/HDL-C ratio (< 1.37) after adjusting for cardiovascular risk factors (OR = 1.83; *p* = 0.03).
Can TG/HDL Ratio be an Accurate Predictor in the Determination of the Risk of Cerebrovascular Events in Youngsters? [63]	Retrospective	Measurements of serum lipids	The TG/HDL-C ratio was significantly higher in young patients with stroke than in healthy adults. Moreover, the TG/HDL-C ratio was also significantly higher in young patients with stroke than in older cases.
The Short-term Prognostic Value of the Triglyceride-to-high-density Lipoprotein Cholesterol Ratio in Acute Ischemic Stroke (AIS) [64]	Retrospective and prospective components	Measurements of serum lipids and review of medical records	The prognosis of the group of patients with a TG/HDL-C ratio > 0.9 was markedly superior to that of the group with a TG/HDL-C ratio ≤ 0.9 (*P* < 0.001). In addition, a higher TG/HDL-C ratio was independently correlated with a reduced risk of mortality (HR = 0.39; *P* < 0.001).
Low triglyceride to high-density lipoprotein cholesterol ratio predicts hemorrhagic transformation in large atherosclerotic infarction of acute ischemic stroke (AIS) [65]	Prospective	Computed tomography or magnetic resonance imaging and blood work	A lower TG/HDL-C ratio was significantly associated with a higher risk of hemorrhagic transformation in patients with AIS attributable to large artery atherosclerosis (OR = 0.53; *p* = 0.032), but not in patients with cardioembolism or small-vessel occlusion.

## Data Availability

Data supporting reported results can be found in the paper’s references. No new data were created.

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
