# Peer review of "The Triglyceride/High-Density Lipoprotein Cholesterol (TG/HDL-C) Ratio as a Risk Marker for Metabolic Syndrome and Cardiovascular Disease"

_diagnostics, 2023, doi:10.3390/diagnostics13050929_

Round 1

Reviewer 1 Report

This manuscript considers some etiological factors for the development of atherosclerotic cardiovascular diseases mainly through three basic groups; coronary artery disease (CAD), peripheral atherosclerosis (PAD) and cerebrovascular disease (CCVD). The authors state that the meaning  Low-density lipoprotein Cholesterol (LDL-C) is mainly regulated by statins, but they state that other lipoprotein components in association with triglycerides can also be an important predictor for the development of atherosclerosis, that is, that the development of atherosclerosis changes in the heart, brain and peripheral blood vessels also depends on transport of other lipids. Elevated triglycerides (TC) in a community with low HDL-C is a proven cause of atherosclerotic cardiovascular disease (ASCVD). Through a very comprehensive review of numerous studies on the TC/HDL-C relationship and TC, results related to metabolic syndrome (MetS), coronary artery disease (CAD), peripheral arterial disease (PAD) and cerebrovascular disease (CCVD) are presented. This proves that the relationship between other lipoproteins in combination, and not just LDL-C levels, is responsible for the development of atherosclerosis. If we look only at this aspect of the development of atherosclerosis, a significant step forward has been made that also includes other diseases such as diabetes and metabolic syndrome. Although inflammation is mentioned at the beginning as part of the atherosclerotic process, perhaps not enough attention is given to it in the paper. Therefore, we think that it would be good if the authors somehow better express the value of inflammatory elements in the development of atherosclerosis, for example in chronic lung disease (proinflammatory cytokines), MIA (metabolic inflammatory atherosclerosis) - atherosclerosis in malnutrition with high values of proinflammatory factors (CRP). This could make the work even more intriguing to the readers. Congratulations on a great effort, we recommend for printing with small changes (additions).

Author Response

The following paragraph was added in the Introduction section (2nd paragraph) to address the reviewer’s comment:

Inflammation plays a crucial role in the pathogenesis of cardiovascular disease (CVD), as it has been linked with both the initiation and progression of atherosclerosis [1,2].  Several pro-inflammatory cytokines, such as the C-reactive protein (CRP), tumor necrosis factor-α (TNF-α) and interleukin-6 (IL-6), have been unequivocally shown to promote both insulin resistance (IR) and atherogenesis [1-3].  Furthermore, inflammation plays a major role in the pathogenesis of chronic lung disease [4], as well as in the pathogenesis of chronic kidney disease (CKD), also including the development of the malnutrition-inflammation-atherosclerosis syndrome (MIA), which is one of the causes of increased mortality in CKD [3,5].

Please note that 5 new references were also added.

Reviewer 2 Report

Summary:

In this review manuscript the authors summarize the knowledge regarding the use of triglyceride and high-density lipoprotein cholesterol ratio to assess the metabolic syndrome and cardiovascular diseases. Overall, this review manuscript discusses a very important topic and gather a significant number of studies reflecting the significance of TG/HDL-C ratio as a risk biomarker.  In addition, the manuscript is very well written and easy to read, but lack the depth of discussion and solid conclusion regarding the recommended cut-of of TG/HDL-C.  

Major:

The discussion or conclusion of this review is very short. For example, the multiple trials have not provided or suggested a clear cut-off value. In fact, the standard deviation among the value provided by this review will undoubtedly be high, which makes it difficult to use GT/HDL-C as a reliable biomarker, even within the same race and gender as some of these studies suggested. Can the authors tackle this issue in depth in their discussion remarks.

Minor:

Line 118-124: confusing paragraph and need to be revised. Please consider the following:

Last but not least, a multicentered study in Brazil, which enrolled 2,472 multiethnic participants free of major cardiovascular risk factors, demonstrated that the defined study TG/HDL-C ratio cut-off values (2.6 for men and 1.7 for women) were reliable and showed promising clinical applicability to detect cardiometabolic disorders. These cut-off values demonstrated great sensitivity and specificity regardless of the ethnicity or age of the participants, although the black race showed low TG/HDL-C ratio values compared with other ethnic groups [21].”

Table 4:  Please correct the typo, “Aa lower TG/HDL-C” to A lower TG/HDL-C”

Author Response

Summary:

 In this review manuscript the authors summarize the knowledge regarding the use of triglyceride and high-density lipoprotein cholesterol ratio to assess the metabolic syndrome and cardiovascular diseases. Overall, this review manuscript discusses a very important topic and gather a significant number of studies reflecting the significance of TG/HDL-C ratio as a risk biomarker.  In addition, the manuscript is very well written and easy to read, but lack the depth of discussion and solid conclusion regarding the recommended cut-of of TG/HDL-C.  

 Major:The discussion or conclusion of this review is very short. For example, the multiple trials have not provided or suggested a clear cut-off value. In fact, the standard deviation among the value provided by this review will undoubtedly be high, which makes it difficult to use GT/HDL-C as a reliable biomarker, even within the same race and gender as some of these studies suggested. Can the authors tackle this issue in depth in their discussion remarks.

This issue was discussed in more depth in the 2nd and 3rd paragraph of the section of “Conclusion” to address the reviewer’s comment.

Please note that 2 new references were also added.

Minor: 

Line 118-124: confusing paragraph and need to be revised. Please consider the following:

 “Last but not least, a multicentered study in Brazil, which enrolled 2,472 multiethnic participants free of major cardiovascular risk factors, demonstrated that the defined study TG/HDL-C ratio cut-off values (2.6 for men and 1.7 for women) were reliable and showed promising clinical applicability to detect cardiometabolic disorders. These cut-off values demonstrated great sensitivity and specificity regardless of the ethnicity or age of the participants, although the black race showed low TG/HDL-C ratio values compared with other ethnic groups [21].”

This paragraph was revised as follows:

Last but not least, a multicentered study in Brazil enrolled 2,472 multiethnic participants free of major cardiovascular risk factors and defined a TG/HDL-C ratio cut-off value of 2.6 for men and 1.7 for women.  The results of this study demonstrated that these cut-off values were reliable and showed good clinical applicability to detect cardiometabolic disorders.  Moreover, these cut-off values demonstrated great sensitivity and specificity regardless of the ethnicity or age of the participants, although the black race showed lower values of TG/HDL-C ratio, compared with other ethnic groups [26].

 Table 4:  Please correct the typo, “Aa lower TG/HDL-C” to A lower TG/HDL-C”

Typo in Table 4 was corrected.

Reviewer 3 Report

In this manuscript, Constantine E. Kosmas et. al. reviewed the current scientific and clinical

data linking the TG/HDL-C ratio with the presence of MetS, CAD, PAD and CCVD. Authots proved the feasibility of TG/HDL-C ratio to be a valuable predictor of above diseases. However, key information of the audiences’ concern is lacking in this manuscript.

Major:

1. The manuscript is more like a reading note than a review. The authors listed relevant study comprehensively, but meanwhile, it is also important to reflect on existing studies and present ideas for future studies.

2. The concentration of lipoproteins has always been a high valued predictor of many diseases. The idea that lipoproteins might be a good predictor of AS had been proposed as early as 1950s (Clough PW. Ann Intern Med. 1957). We already receive the concept that lipoprotein levels might be important in predicting diseases. However, as the authors mentioned in the Conclusion, the absence of a universal cut-off value renders its use as a predicting biomarker for CVD challenging. The audiences are more interested in specific, quantitative methods of prediction, and less interested in the list of existing literature

Minor:

1. Up to date, it is still controversial that which combination of lipoproteins can predict certain diseases best. The combinations are further complicated with the addition of new parameters such as non-traditional lipid parameters, disease-specific parameters, and inflammation-related parameters. Considering that there have been so many heterogeneous studies, it is necessary to compare various combinations of predictors, and state the reason why TG/HDL-C is better than other combinations (rather than stating that TG/HDL-C has advantages).

2. The number of Conclusion is supposed to be 6.

Author Response

Reviewer 3

Comments and Suggestions for Authors

In this manuscript, Constantine E. Kosmas et. al. reviewed the current scientific and clinical data linking the TG/HDL-C ratio with the presence of MetS, CAD, PAD and CCVD. Authors proved the feasibility of TG/HDL-C ratio to be a valuable predictor of above diseases. However, key information of the audiences’ concern is lacking in this manuscript.

Major:

  1. The manuscript is more like a reading note than a review. The authors listed relevant study comprehensively, but meanwhile, it is also important to reflect on existing studies and present ideas for future studies.

We believe that our article has all the characteristics of a thorough “review article”, as we provide the reader with a valuable, solid, informative, critical summary of our topic.  In our article, we summarize the current state of understanding on the topic of “the TG/HDL-C ratio as a Risk Marker for MetS and CVD” by presenting and discussing the data available from existing studies.  Actually, we have made every effort to provide data from the most current state of the literature, as more than 50% of the references discussed in our review have been published within the last 4 years.  Thus, we feel that our review article on “The TG/HDL-C ratio as a Risk Marker for MetS and CVD” (if published) would be a very informative update on the subject and the readers would get a very good grasp on the topic.

However, to fully address the reviewer’s comment, in our revised article we have included five (5) ideas/proposals for future studies, exactly as requested by the reviewer, as follows:

  1. At the end of the 8th paragraph of the section “Triglyceride/HDL-C ratio in Coronary Artery Disease”:

“Notwithstanding, further studies are needed to definitely assess the role of TG/HDL-C ratio as a risk marker for ISR.”

  1. At the end of the 2nd paragraph of the section “Triglyceride/HDL-C ratio in Peripheral Artery Disease”:

“Thus, further studies are needed in order to establish clinically reliable risk markers that would predict the risk for PAD.”

  1. 9th paragraph of the section “Triglyceride/HDL-C ratio in Cerebrovascular Disease”:

“Nonetheless, further large studies are needed to definitely assess the role of the TG/HDL-C ratio as a predicting marker of the risk of stroke and its prognosis.”

  1. 4th paragraph of the section of “Conclusion”:

“The above data ought to set the foundation for future exploring the best possible cut-off values, sensitivity and specificity of the TG/HDL-C ratio across diverse racial/ethnic subgroups, so as to add this ratio more effectively in the diagnostic armamentarium of CVD.”

  1. At the beginning of the 5th paragraph of the section of “Conclusion”:

“In addition, further studies are needed to establish other new lipid-related biomarkers and to compare the reliability of these biomarkers, or even different combinations of them, in terms of their clinical efficacy in predicting the risk for MetS and CVD.”

  1. The concentration of lipoproteins has always been a high valued predictor of many diseases. The idea that lipoproteins might be a good predictor of AS had been proposed as early as 1950s (Clough PW. Ann Intern Med. 1957). We already receive the concept that lipoprotein levels might be important in predicting diseases. However, as the authors mentioned in the Conclusion, the absence of a universal cut-off value renders its use as a predicting biomarker for CVD challenging. The audiences are more interested in specific, quantitative methods of prediction, and less interested in the list of existing literature.

To address the reviewer’s comment, this issue was discussed in more depth in the 2nd and 3rd paragraph of the section of “Conclusion”, where specific quantitative methods of prediction used in certain large studies (receiver operating characteristic [ROC] analysis, binomial regression analysis, etc.) were presented.

Minor:

  1. Up to date, it is still controversial that which combination of lipoproteins can predict certain diseases best. The combinations are further complicated with the addition of new parameters such as non-traditional lipid parameters, disease-specific parameters, and inflammation-related parameters. Considering that there have been so many heterogeneous studies, it is necessary to compare various combinations of predictors, and state the reason why TG/HDL-C is better than other combinations (rather than stating that TG/HDL-C has advantages).

To address the reviewer’s comment, the following paragraphs were included in the revised manuscript (last 2 paragraphs of the section of “Conclusion” in the revised manuscript):

“In addition, further studies are needed to establish other new lipid-related biomarkers and to compare the reliability of these biomarkers, or even different combinations of them, in terms of their clinical efficacy in predicting the risk for MetS and CVD.  Moreover, relevant comparison of the new lipid-related biomarkers with the TG/HDL-C ratio would be important.  This may also be useful for establishing guidelines for the prevention of CVD in clinical practice.

Notwithstanding, the above should not take away the fact that the TG/HLC-C ratio has been proven to be an excellent novel risk marker for effectively predicting the risk for MetS and CVD.

  1. The number of Conclusion is supposed to be 6.

Number was changed to 6.

Round 2

Reviewer 3 Report

The authors didn't address my comments.